# Automated Graphic Divergent Thinking Assessment: A Multimodal Machine Learning Approach

**DOI:** 10.3390/jintelligence13040045

**Published:** 2025-04-07

**Authors:** Hezhi Zhang, Hang Dong, Ying Wang, Xinyu Zhang, Fan Yu, Bailin Ren, Jianping Xu

**Affiliations:** 1Faculty of Psychology, Beijing Normal University, Beijing 100875, China; psy_zhanghz@mail.bnu.edu.cn (H.Z.); donghang2338@gmail.com (H.D.); 202228061280@mail.bnu.edu.cn (Y.W.); zxy201811061186@gmail.com (X.Z.); 202328061277@mail.bnu.edu.cn (F.Y.); bailinn0818@gmail.com (B.R.); 2Faculty of Arts and Sciences, Beijing Normal University, Zhuhai 519085, China

**Keywords:** divergent thinking, graphic completion, automated scoring, deep learning, multimodality

## Abstract

This study proposes a multimodal deep learning model for automated scoring of image-based divergent thinking tests, integrating visual and semantic features to improve assessment objectivity and efficiency. Utilizing 708 Chinese high school students’ responses from validated tests, we developed a system combining pretrained ResNet50 (image features) and GloVe (text embeddings), fused through a fully connected neural network with MSE loss and Adam optimization. The training set (603 images, triple-rated consensus scores) showed strong alignment with human scores (Pearson r = 0.810). Validation on 100 images demonstrated generalization capacity (r = 0.561), while participant-level analysis achieved 0.602 correlation with total human scores. Results indicate multimodal integration effectively captures divergent thinking dimensions, enabling simultaneous evaluation of novelty, fluency, and flexibility. This approach reduces manual scoring subjectivity, streamlines assessment processes, and maintains cost-effectiveness while preserving psychometric rigor. The findings advance automated cognitive evaluation methodologies by demonstrating the complementary value of visual-textual feature fusion in creativity assessment.

## 1. Introduction

In the era of the knowledge economy, creativity is undoubtedly the core force driving continuous social progress and economic prosperity, encompassing various fields such as science, technology, and the arts ([27]). The assessment of creativity has been evolving for a considerable time ([17]), with divergent thinking—the cognitive ability to generate multiple creative ideas in response to a specific problem—considered a significant manifestation of creativity ([1]). Precise evaluation and effective screening of individual creativity have become key issues of common interest in both academia and practice.

The evaluation of divergent thinking outcomes has always been a focus of attention. Traditional scoring methods often include fluency, flexibility, and originality, with assessments of flexibility and originality often based on subjectivity or experience ([16]; [31]; [36]). This framework is plagued by subjectivity; different raters, owing to their professional backgrounds, cognitive preferences, and experience can lead to insufficient scoring accuracy. The statistical scoring process also consumes a significant amount of human resources and is time-consuming. Therefore, it is necessary to develop a rapid and effective scoring method. These issues have been partially resolved by advances in computer science. In recent years, with the rapid development of artificial intelligence technology, especially the widespread application of deep learning in the field of natural language processing, it has become possible to a large extent to mimic human cognitive abilities to perform corresponding tasks. AI algorithms can assist in almost every aspect of psychometrics, providing new ideas and methods for the automated scoring of divergent thinking tests ([10]). Machine learning-based scoring methods can reduce the time required for measurement ([23]) to some extent, reducing the bias in manual scoring ([39]). However, there is still a lack of research on combining images and their meanings in divergent thinking tests involving images.

### 1.1. Divergent Thinking Tests

Divergent thinking, a key component of creative thinking, has been a topic of interest in psychology for many years. Traditionally, divergent thinking (DT) has been widely regarded as a core dimension of creativity. However, Runco ([32]) emphasized that DT is not synonymous with creativity, arguing that creativity necessitates the integration of convergent thinking and sociocultural value. He called for the maintenance of objectivity in assessment practices to distinguish between these constructs. This perspective establishes a theoretical foundation for subsequent research, framing a critical question: how can DT be measured effectively and precisely to advance creativity research?

The Torrance Tests of Creative Thinking (TTCTs), rooted in Guilford’s model ([37]), have demonstrated statistically significant correlations with creative achievement in longitudinal studies spanning 9, 7, 22, and 40 years ([13]; [22]; [38]). Specifically, Torrance’s ([38]) 22-year longitudinal study and reanalyses by [40] ([40]) and [28] ([28]) of Torrance’s data have concluded that the Creative Index is the most robust predictor of adult creative achievement. TTCTs have long served as the gold standard for DT assessment ([19]).

While critiques of the Torrance Tests of Creative Thinking (TTCTs; [7]) prompted renewed interest in alternative measures, simplified tasks, such as the Alternative Uses Task (AUT), have previously been examined for their feasibility in large-scale contexts. In [33]’s ([33]) study, participants were instructed to generate creative uses for bricks and later circled their two most creative responses. Raters then evaluated these responses on a 5-point scale, demonstrating that subjective ratings of unusual use tasks yielded reliable and consistent scores with only two or three raters. This approach highlighted the reliability and validity of the AUT as a measure of divergent thinking, showing that it could effectively assess creativity in a way that was less confounded by fluency and more independent of the sample size.

For example, the figural subtest asks individuals to make abstract line drawings into meaningful images, with scoring criteria spanning fluency, originality, and elaboration ([37]). This approach aligns with [17]’s ([17]) emphasis on non-verbal ideation as a core component of DT. The TTCT-figural additionally evaluates resistance to premature closure and abstractness of titles. The latter criterion underscores the interplay between figural and verbal components in creativity assessment, a rationale that informed our multimodal approach. Recent adaptations include combination tasks that integrate verbal and figural elements such as constructing narratives from visual prompts or synthesizing abstract symbols into coherent stories ([4]).

The reliability and validity of the scoring methods directly determine the utility of the DT assessments. Plucker systematically examined the risks of subjective bias in DT scoring, advocating for hybrid approaches that combine automated metrics (e.g., fluency and originality) with expert ratings to enhance validity. Automated metrics for originality typically rely on response rarity (e.g., statistical uniqueness), whereas expert ratings incorporate subjective judgments of novelty and appropriateness ([30]). Subsequent studies refined these methods by introducing semantic association distance to quantify idea novelty, revealing that DT’s cognitive essence lies in its ability to form remote associations ([1]). [29] ([29]) validated the necessity of multidimensional scoring (e.g., practicality and elaboration) in real-world tasks, underscoring the importance of contextualized assessment.

Plucker emphasized that advancements in DT assessment directly inform the identification and nurturing of student creativity in educational settings ([30]). Emerging trends suggest that integrating neuroscience (e.g., neural network flexibility) and computational linguistics (e.g., semantic network analysis) may transcend traditional scoring limitations, although balancing ecological validity with standardization remains a critical challenge ([3]).

### 1.2. DT Scoring Method

Divergent thinking tests typically assess three dimensions of the respondents’ answers: fluency, flexibility, and originality. Academic consensus has been reached on the scoring criteria for fluency and flexibility: the former is based on counting the number of all reasonable and logical answers that can be used for a familiar item, whereas the latter is determined on the basis of the number of conceptual categories or the number of switches between conceptual categories. ([34]; [2]; [21]). However, there is no consensus in academia regarding a scoring method for the originality dimension.

Traditional scoring methods for originality in divergent thinking tests typically assign one point to unique answers (those given by only one participant) and zero points to non-unique answers, reflecting creativity’s originality ([33]). [9] ([9]) proposed a subjective top-scoring method where participants self-select their most creative ideas. However, this approach demonstrated high concordance with external ratings. Other approaches included using a scoring key list of non-unique answers or awarding 1 point to answers given by less than 5% of the sample, with all others receiving 0 points. These methods focus on accumulating points for unique responses in order to assess originality. Reiter ([31]) argued that originality scoring should go beyond traditional approaches. Frequency-based scoring measures response rarity objectively but requires subjective judgments for response similarity. Rater-based scoring captures additional nuances of originality but is prone to subjectivity. Combining multiple indicators provides a more comprehensive assessment, albeit with an increased complexity. Other methods, such as old/new response scoring, ideational pool scoring, top two scoring, appropriateness scoring, and category switching scoring, offer unique insights, but each has limitations.

However, these scoring methods have several challenges. First, there is a high correlation between fluency and originality scores, which may lead to confusion between the two concepts ([12]; [33]). Through latent variable analysis, [33] ([33]) found that the correlation coefficient between fluency and originality was extremely high (r = 0.88), making it difficult for originality scores to be independent of fluency scores and to explain other variables on their own. Second, originality scores are highly sensitive to sample size; changes in sample size can render answers that were once considered unique as no longer unique ([24]). Third, frequency-based originality scoring is highly sample-dependent, with insufficient reliability in small samples. A sample size of ≥300 was required to achieve higher reliability. Additionally, the measurement error is the greatest for unique responses, potentially underestimating the performance of highly creative individuals. It is necessary to adjust the sample sizes or methods based on task characteristics such as the solution space ([16]). Finally, for divergent thinking tests involving image completion, there is also the issue of respondents’ drawing abilities and raters’ interpretations of the drawings, which can introduce significant errors.

### 1.3. Deep Learning and Multimodality

Deep learning is a significant branch within the field of machine learning, centered around constructing multilayered neural network models that automatically extract high-level feature representations through extensive data training. This multilayered structure enables machines to mimic the information processing of the human brain, mapping low-level signals to high-level semantics. The origins of deep learning can be traced back to neural network research in the 1980s, but it gained widespread attention in 2006 when Hinton and colleagues introduced the Deep Belief Network (DBN) model. Image recognition is one of the most widely applied areas of deep learning. Convolutional neural networks (CNNs) have demonstrated exceptional performance in tasks such as image classification, object detection, and image segmentation. For instance, AlexNet achieved a breakthrough in the 2012 ImageNet competition, which greatly propelled the development of deep learning in the field of image recognition.

Multimodal technology enhances a machine’s ability to understand and process complex environments and information by integrating data from different modalities such as text, images, and audio. The key challenge in multimodal technology is effectively representing and integrating data from different modalities and establishing contextual relationships between them. In divergent thinking tests involving image completion, the responses of the participants, in addition to the drawn figures, reflect the divergence of their thinking through their definitions of the figures, making semantics one of the criteria for evaluation.

Technological advancements have provided new ideas and methods for the automated scoring of divergent thinking tests. Researchers have begun to apply deep learning models to the processing and classification of response texts, aiming to simplify the complex scoring process into an efficient text-classification task. The core of this method is to train deep learning models to recognize and distinguish responses of varying qualities or categories, thereby automatically assigning corresponding scores. Simultaneously, to reduce the subjective influence of manual scoring, some researchers have used semantic distance calculations to standardize the performance of divergent thinking tests, addressing the subjectivity of traditional scoring methods ([35]; [8]). This approach employs large language model-supervised learning methods, which show a high correlation with human scoring (r = 0.80), closely approximating human assessment. Computational linguistics has also been used to characterize the salient features of creative texts, expanding previous work on the automated scoring of divergent thinking. Latent Semantic Analysis (LSA) has been utilized to score brief responses measuring creativity and divergent thinking. Furthermore, various supervised learning methods have been compared for the automated scoring of divergent thinking tasks, indicating a shift towards machine-based scoring methods ([20]; [11]). And it can effectively reduce the time required for the manual scoring of a large number of answers in the test and save on the cost of a certain test. These findings demonstrate the superiority of supervised learning methods and large language models in scoring language-based tests ([25]).

Significant advancements have been made in the field of automated scoring for figural divergent thinking (DT) tests in recent years. Early research primarily focused on applying automated scoring to figural creativity tests such as the Torrance Tests of Creative Thinking-Figural (TTCT-F) and the Multi-Trial Creative Ideation task (MTCI). [14] ([14]) first attempted to use convolutional neural networks (CNNs) for automated scoring of the Test of Creative Thinking–Drawing Production (TCT-DP), achieving an accuracy comparable to human raters (approximately 90%) on a limited test dataset. However, the small sample size restricts the generalizability of the findings. [15] ([15]) further developed a large-scale image classification model for TCT-DP, achieving an average accuracy of 80.4% and a Pearson correlation coefficient of r = 0.81 with human ratings, demonstrating the potential of machine learning algorithms to address the fitness-for-purpose problem in creativity measurement, highlighting the promise of automated scoring methods for figural creativity tests, offering rapid, accurate, and cost-effective assessments that can be seamlessly integrated into teaching and learning environments.

Subsequently, [26] ([26]) developed the Automated Drawing Assessment (AuDrA) platform, which utilizes a deep convolutional neural network (ResNet) to score simple line drawings from the MTCI task. AuDrA demonstrated remarkable performance on a training dataset of over 13,000 drawings, with correlations as high as 0.76 between its scores and human raters’ judgments. The platform also exhibited strong generalizability across different datasets, highlighting its sensitivity to drawing features beyond mere complexity ([26]).

In another study, [5] ([5]) explored automated scoring methods for TTCT-F and MTCI. They employed Random Forest classifiers on TTCT-F, achieving an accuracy of up to 85%, and further enhanced the scoring accuracy by incorporating both image and title information. Additionally, they tested Vision Transformer models such as BEiT and ViT, which demonstrated higher correlations (r = 0.85) on the MTCI dataset, indicating the superior potential of Vision Transformers in handling figural creativity tests ([5]).

Based on the above, we posed the following questions: Is it effective to evaluate divergent thinking tests involving image completion by combining scores for figures and their descriptive meanings? Can deep learning and multimodal methods be utilized to score image completion-based divergent thinking tests and can these scores effectively reflect the participants’ levels of divergent thinking? Can the models score consistently and possess a certain degree of transferability? While prior work has focused on unimodal scoring (e.g., image differences or text alone), our study uniquely integrates image and textual descriptions via deep learning to capture multimodal creativity signals. This approach addresses limitations of existing methods, such as overlooking the semantic context in purely visual models or neglecting graphic originality in text-based systems.

## 2. Methods

### 2.1. Design and Participants

First, data were collected using a well-validated self-designed “High School Students’ Divergent Thinking Test”, which includes both verbal and drawing tasks and is designed to examine students’ fluency, flexibility, and novelty. In this test, only the novelty scores were used for machine training. We focused on novelty (rather than originality) because each drawing task was inherently original, and we aimed to assess whether the students generated unusual ideas within the same conceptual category. Novelty reflects both uniqueness and unexpectedness, aligning with the goal of evaluating creative thinking.

A total of 708 Chinese high school sophomores participated in the high stakes test, with physics-majored students accounting for 70.3% of the participants. Ultimately, 708 datasets were retained. The drawing tasks were uniformly scanned, and unique identifiers were assigned. The test required participants to complete a meaningful drawing based on a regular pentagon within a limited time and label its meaning alongside it (Figure 1). Written informed consent was obtained from all participants before the experiment, and the study protocol was approved by the Institutional Review Board of the Faculty of Psychology at Beijing Normal University (Protocol ID: BNU202301230217).

Second, the drawing tasks were segmented into nine uniform-sized images using the PIL (Pillow) library in Python 3.10, resulting in a dataset comprising 6372 individual images for subsequent model training (Figure 2).

### 2.2. Annotation

To achieve high-quality semantic labeling, six graduate students majoring in psychology were invited to complete meaning annotation for each image, as shown in Figure 3. Because the test required participants to provide descriptions of their drawings, these descriptions were referenced during the annotation process. Before formal annotation, a preliminary round of annotation was conducted, followed by discussions to establish unified standards for labeling, particularly defining what constitutes abstract or meaningless lines.

### 2.3. Manual Scoring

Six graduate students specializing in psychology rated the novelty of each image on a scale of 0 to 3 using a validated scoring rubric; these ratings were used for subsequent machine learning. Initially, 36 responses were scored to establish the consistency of manual scoring.

The dataset for training comprised responses from 67 participants, with a total of 603 images. Each image was scored by three trained raters. To minimize subjective differences, the final score for each image was determined based on the ratings from the three raters as follows:If all three raters agreed on the score, that score was taken as the image’s rating.If two out of three raters were in agreement, and the third rater’s score differed by more than 2 points from the other two (e.g., 0, 0, 3), the score of the two raters who agreed was taken as the image’s rating.If the three raters all disagreed or the differences were minor, the average score of the three raters was taken as the novelty score.

### 2.4. Model Construction

#### 2.4.1. Feature Extraction and Fusion

##### Text Feature Extraction

For the text-based model (TEXT), annotated responses were first tokenized using the Jieba segmentation tool. The tokenized words were then converted into 100-dimensional word vectors using the GloVe model pretrained on the Gigaword corpus. The feature representation for each text input was derived by averaging all the word vectors in the corresponding annotation.

##### Image Feature Extraction

For the image-based model (IMG), a pretrained ResNet50 convolutional neural network (initialized with ImageNet weights) was employed to extract visual features. To enhance generalization, data augmentation techniques, including rotation (±20°), horizontal and vertical shifts (±20% of image dimensions), horizontal flipping, zooming (±20%), shearing (±0.2 rad), and nearest-neighbor padding, were applied to the input images. The augmented images were fed into ResNet50, and the features from the global average pooling layer (after removing the fully connected layers) were extracted as image representations.

##### Multimodal Feature Fusion

For the combined model (comb), the text and image feature vectors (100-dimensional and 2048-dimensional, respectively) were concatenated into a unified 2148-dimensional vector, integrating both semantic and visual information.

#### 2.4.2. Model Training

Three distinct fully connected neural networks were constructed: TEXT (text-only), IMG (image-only), and comb (multimodal). Each model comprised two hidden layers (128 and 64 neurons) with ReLU activation, followed by a single-neuron output layer for regression. Mean squared error (MSE) served as the loss function, and the Adam optimizer was used for training.

The modeling dataset (603 images) was randomly chosen from the entire database and partitioned into training and validation sets in an 8:2 ratio. For robust evaluation, each model was trained with varying epochs (10–40, in increments of 2), and three independent training runs were conducted per epoch configuration, resulting in 45 candidate models per model type. During training, the batch data were dynamically generated using a generator to prevent overfitting. The training and validation loss curves were monitored to ensure model stability. Model selection prioritizes minimizing the validation loss (MSE) and maximizing the Pearson correlation with human scores.

#### 2.4.3. Model Validation

Additionally, three raters scored a randomly selected subset of 100 responses, and the machine scoring system also provided scores. The Pearson correlation between the machine and human scores was calculated to evaluate the effectiveness and generalization capabilities of the model. Furthermore, the total scores for each participant on the task were compared with machine-generated totals to assess the model’s usability.

## 3. Results

### 3.1. Inter-Rater Reliability and Scoring Scheme Validity

The Fleiss’ Kappa for multiple raters was initially calculated to be 0.378, indicating a moderate level of agreement among the raters. The following discussions and revisions to the scoring rubric and Fleiss’ Kappa for the same 200 data points were recalculated and found to be greater than 0.6, which signifies a higher degree of inter-rater reliability. This improvement suggests that the scoring scheme was refined to achieve better consistency among the raters, thereby validating its effectiveness.

### 3.2. Automated Scoring Model Performance

The optimal models for each modality were selected based on the validation loss (MSE) and Pearson correlation with human scores. During training, each model configuration (TEXT, IMG, and COMB) was independently trained three times across varying epochs (10 to 40, in increments of 2), resulting in 45 candidate models per modality. Training progress was monitored by tracking the validation loss and Pearson’s correlation over the epochs. When the epochs were greater than 30, both metrics stabilized across all models, indicating convergence. The final models were chosen by prioritizing the minimal validation loss and maximal Pearson correlation for the validation set. Figure 4, Figure 5 and Figure 6 illustrate the training dynamics of each modality.

The selected models demonstrated the following performances on the training data (Std. refers to the standard deviation across the three independent training runs per epoch configuration).
TEXT: text_model_epochs32_v1.keras achieved an MSE of 0.428 and a Pearson correlation of r = 0.910.IMG: img_model_epochs34_v0.keras yielded an MSE of 0.558 and r = 0.915.COMB: comb_model_epochs36_v2.keras outperformed others with an MSE of 0.350 and r = 0.946.

### 3.3. Comparison of Automated Scoring with Human Rating

To evaluate the model performance at both the individual response and participant levels, Pearson correlations were calculated between the machine-predicted scores and human ratings. On the original training dataset (603 images), the text-only model (TEXT) achieved a high correlation of r = 0.910 for individual responses, whereas the image-only model (IMG) showed a moderate correlation (r = 0.700). The combined multimodal model (comb) achieved r = 0.810. When tested on an independent validation set of 100 images, the correlations decreased but maintained meaningful predictive validity: TEXT (r = 0.629), IMG (r = 0.395), and comb (r = 0.561) (Figure 7, Figure 8, Figure 9 and Figure 10).

At the participant level, the total scores (summing nine responses per participant) were analyzed. In the validation set, the TEXT model demonstrated strong generalization (r = 0.724 with human total scores), whereas the IMG model exhibited a weak alignment (r = 0.135). The comb model achieved a correlation of r = 0.602, indicating that integrating multimodal features partially preserved the predictive power despite the inherent challenges of cross-domain generalization (Figure 11 and Figure 12).

## 4. Discussion

### 4.1. Model Performance Analysis and the Advantages of Multimodality

#### 4.1.1. Enhancing Novelty Scoring Through Semantic Features

The inclusion of semantic annotations significantly improved the validity of automated scoring compared to purely visual models. While the IMG model relied solely on image features, which may capture abstract visual patterns but lack an explicit semantic context, the TEXT model achieved higher correlations (r = 0.910 vs. r = 0.700 on training data). It should be noted that these results were based on the entire training dataset and may not directly indicate the superiority of the multimodal model over other models. The high correlation of the text-only model is likely due to the nature of the existing scoring method, which is based on the frequency statistics of human ratings of drawing types. This aligns with the manual scoring rubric, where raters explicitly referenced textual labels to interpret drawings. However, the comb model’s intermediate performance (r = 0.810) suggests that multimodal integration introduces complementary strengths: visual features capture graphical originality, whereas text features contextualize symbolic meaning.

Critically, the IMG model’s poor generalization (r = 0.135 for total scores) highlights the limitations of relying solely on visual input for tasks requiring semantic interpretation. Divergent thinking involves both ideational fluency (quantified by text) and graphic originality (captured by images). Thus, while semantic annotations may dominate training correlations owing to their alignment with manual scoring criteria, the multimodal approach better reflects the holistic nature of creativity assessment. Future studies should explore hybrid frameworks that dynamically weigh visual and semantic features based on task requirements.

#### 4.1.2. Strengths and Challenges of Multimodality

Our findings underscore the necessity of balancing visual and semantic inputs. For instance, the comb model’s validation performance (r = 0.602 on total scores) outperformed the IMG model but fell short of that of the TEXT model. This discrepancy may arise because textual labels directly encode raters’ subjective interpretations, whereas image features require deeper abstraction to align with human judgments. Nevertheless, the multimodal framework offers a more ecologically valid representation of divergent thinking, as real-world creativity often manifests through both visual and verbal channels.

To further validate this approach, future work could compare models trained exclusively on images (without labels) with those trained using multimodal inputs. Such experiments would clarify whether semantic annotations genuinely enhance scoring validity or merely replicate rater bias. Additionally, incorporating advanced techniques, such as attention mechanisms, could optimize feature fusion, enabling models to prioritize contextually relevant modalities during prediction.

### 4.2. Theoretical and Practical Significance

The integration of psychometrics and computer science enhances the efficiency and effectiveness of measurements, reduces human bias ([6]), and improves the quality of assessment and scoring ([18]). By combining multimodal deep learning models of image and text features, this study contributes to the automation of divergent thinking assessment research. This demonstrates how to effectively extract visual and textual features using pretrained models and successfully integrate these features, providing new insights for feature integration in complex multimodal tasks. Not only does it explore the potential for automating the assessment of graphical divergent thinking but it also offers a viable framework for the automation of divergent thinking scoring based on large language models, providing suggestions for future research in divergent thinking.

Furthermore, this study integrated scoring criteria for divergent thinking. By utilizing convolutional neural networks (CNNs) and word vector models for a comprehensive analysis of novelty, this study integrates fluency and flexibility into the measurement of novelty, reducing the complexity of multidimensional measurement. This integrative approach can help subsequent researchers delve deeper into interactive mechanisms and dynamic changes in the process of innovative thinking.

Finally, this study also provides a reference for the evaluation systems of other innovative thinking abilities. Future research can extend to other areas of innovative thinking such as professional creativity and design thinking. By drawing on the methods of this study, researchers can develop corresponding automated scoring systems to analyze and assess different types of innovative abilities, providing objective and reliable assessment results. Simplifying the assessment process achieves an efficient evaluation. With automated scoring systems, raters can quickly and accurately assess the creative thinking levels of participants, thereby enabling them to better understand their capabilities.

In addition, the cost is reduced, and its application is widely expanded. Automated scoring systems can significantly reduce these costs, allowing assessments to be conducted without incurring additional expenses, facilitating broader application and promotion.

### 4.3. Limitations and Prospects

Although the automated scoring method for divergent thinking tests has shown some initial promise, there are still several limitations that need to be further explored and addressed in future research.

First, in the automatic annotation process of image responses, the accuracy of OCR and AI recognition technologies is still not ideal, which means that current automated scoring methods still rely on manual annotation to ensure prediction effectiveness, limiting the promotion and application of automated scoring methods. This may be because of the difficulty of recognizing handwritten texts. In future test administrations, it may be possible to optimize the answer sheet design by fixing the number of words and response areas for candidates’ self-naming of drawings, thereby facilitating subsequent automated recognition. Moreover, divergent thinking, as an abstract trait, may have strong specificity in drawing and textual naming. Future research should attempt to introduce larger language models or explore unsupervised learning methods to reduce reliance on manually annotated data and enhance the model’s generalization capabilities.

Second, the manual scoring strategy must be improved. A linear prediction model was used in this study. The manual scoring criteria for the drawing questions in the “High School Students’ Divergent Thinking Test” included three dimensions, fluency, flexibility, and novelty, which is consistent with past research on the dimensions of divergent thinking. After expert evaluation, the one-dimensional automated scoring results obtained in this study can cover these three dimensions. In the future, classification prediction models could be considered to more closely integrate traditional divergent thinking dimensions, predicting fluency, flexibility, and novelty as separate categories.

Third, the internal mechanisms of models trained by deep learning methods have a “black-box” problem, limiting their interpretability, and the training results have a certain degree of uncertainty. Future research could use visualization techniques to make the model’s internal features more intuitive, helping to understand the model’s decision-making process. At the same time, exploring more interpretable model structures, such as decision trees or rule systems, or testing the model’s robustness and reliability through adversarial sample tests can further enhance the model’s credibility.

Lastly, considering that the participants in this study were all high school students, future research can expand the sample range to include participants of different age groups and educational levels and train models with a broader dataset to enhance the universality and applicability of automated scoring methods.

## 5. Conclusions

This study developed an automated scoring method for divergent thinking tests based on multimodal deep learning models by integrating both image and text features. The results indicate that the automated scoring method achieves a Pearson correlation of 0.810 with the human scoring results on the training dataset, demonstrating a high level of correlation between machine ratings and human ratings. Importantly, on a separate validation dataset that was not used in the training, the model achieved a Pearson correlation of 0.561, indicating that the model possesses a certain degree of generalization ability. This approach not only enhances the efficiency and objectivity of scoring but also provides a more comprehensive and holistic assessment of divergent thinking by combining visual and semantic features. These findings highlight the importance of multimodal integration in creativity assessment and suggest that combining visual and semantic features can better capture the multifaceted nature of divergent thinking than single-modality approaches.

## Figures and Tables

**Figure 1 jintelligence-13-00045-f001:**
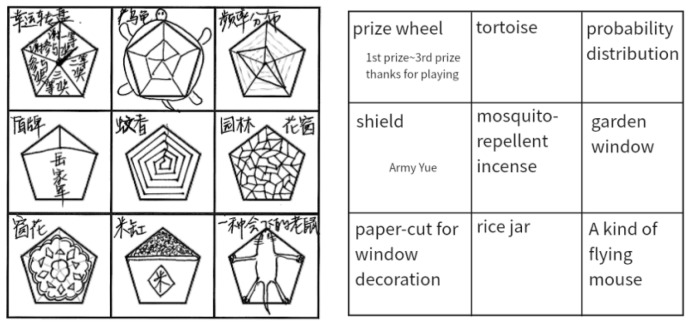
A sample response.

**Figure 2 jintelligence-13-00045-f002:**
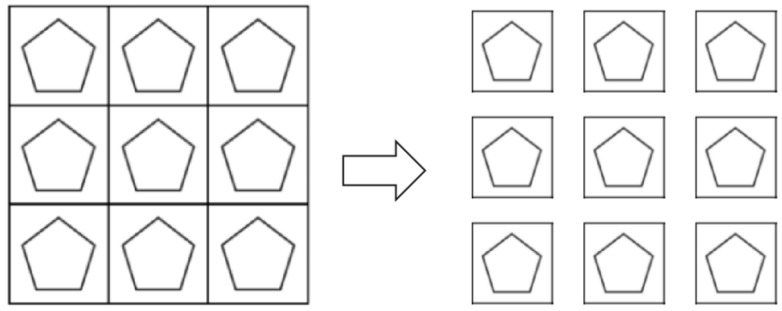
Image cropping method.

**Figure 3 jintelligence-13-00045-f003:**
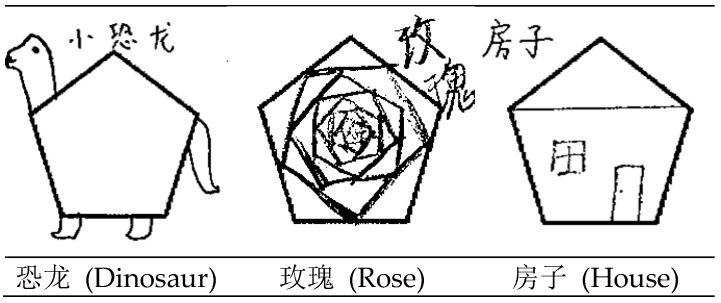
Example of image annotation.

**Figure 4 jintelligence-13-00045-f004:**
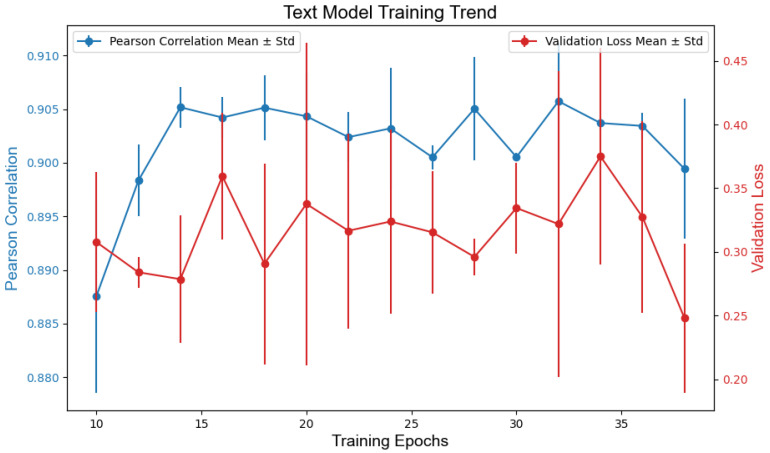
Model training loss and correlation based on text.

**Figure 5 jintelligence-13-00045-f005:**
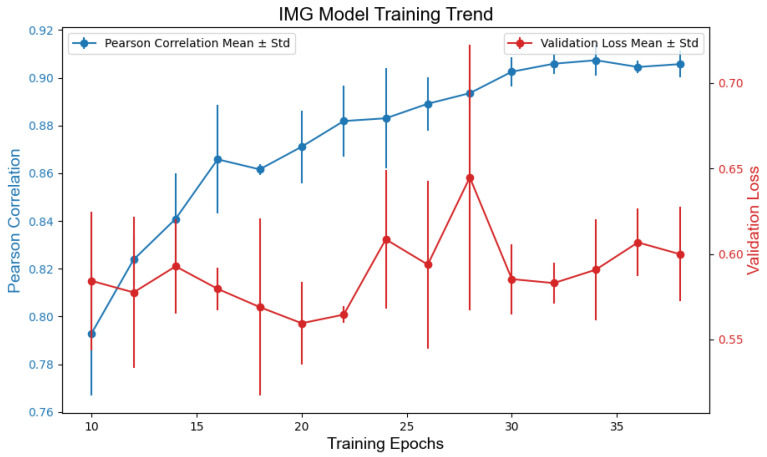
Model training loss and correlation based on image.

**Figure 6 jintelligence-13-00045-f006:**
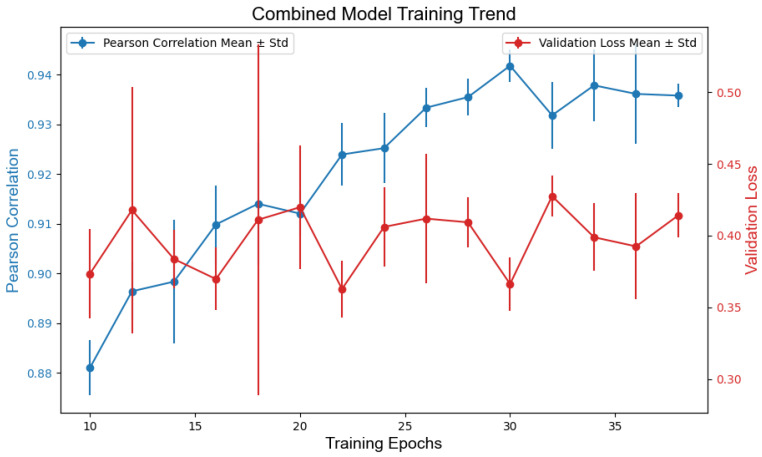
Model training loss and correlation based on combined features.

**Figure 7 jintelligence-13-00045-f007:**
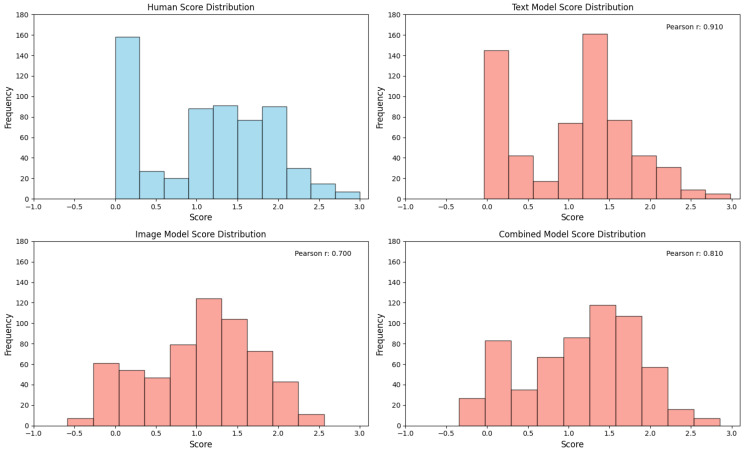
The score distribution on dataset 603 under different scoring methods. **Top-left**: Distribution of human expert ratings (ground truth). **Top-right**: Prediction distribution of the text-only model (text_model). **Bottom-left**: Prediction distribution of the image-only model (img_model). **Bottom-right**: Prediction distribution of the multimodal fusion model (comb_model).

**Figure 8 jintelligence-13-00045-f008:**
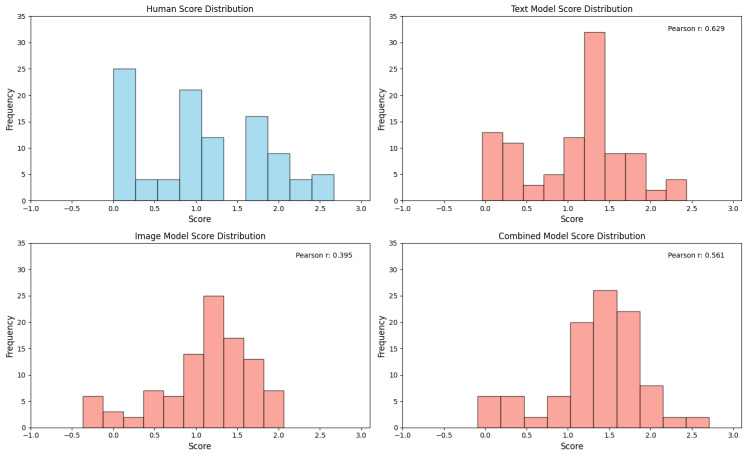
The score distribution on dataset 100 under different scoring methods. **Top-left**: Distribution of human expert ratings (ground truth). **Top-right**: Prediction distribution of the text-only model (text_model). **Bottom-left**: Prediction distribution of the image-only model (img_model). **Bottom-right**: Prediction distribution of the multimodal fusion model (comb_model).

**Figure 9 jintelligence-13-00045-f009:**
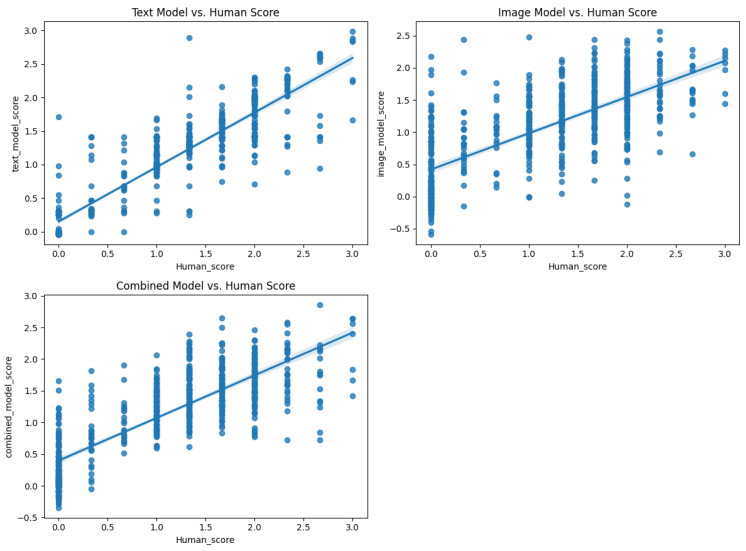
Scatter plot of different model scores against human scores (603 database).

**Figure 10 jintelligence-13-00045-f010:**
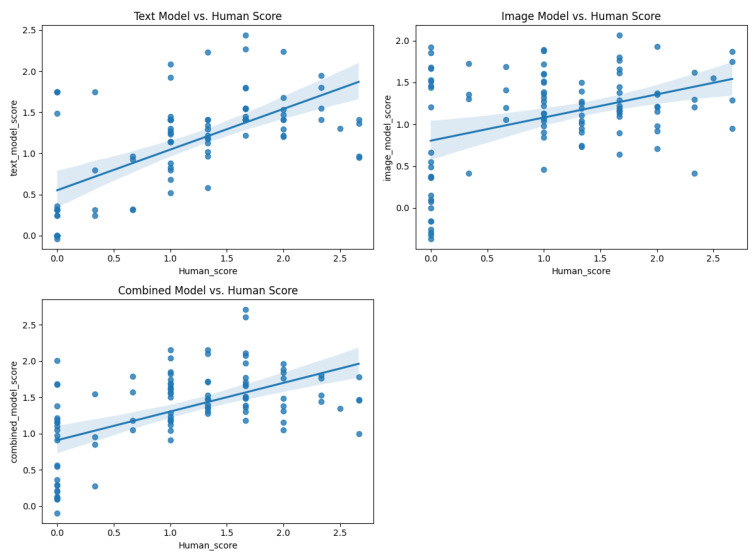
Scatter plot of different model scores against human scores (100 database).

**Figure 11 jintelligence-13-00045-f011:**
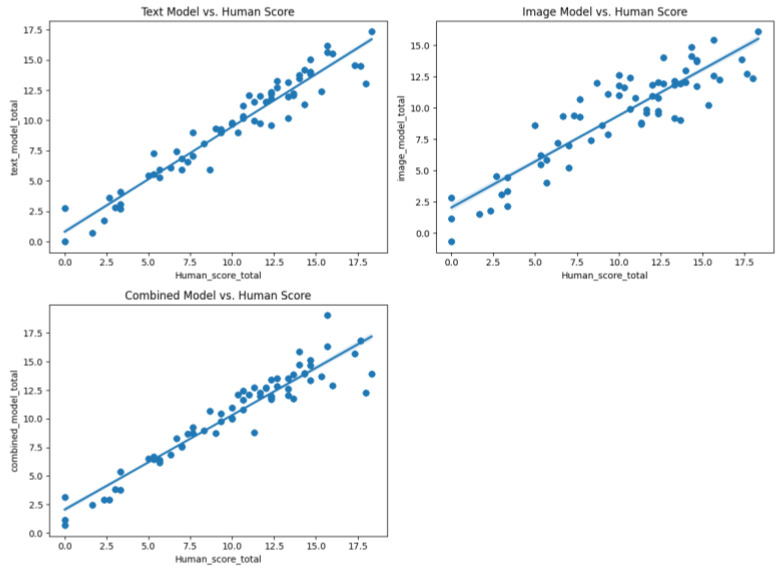
Scatter plot of different model sum scores against human scores (603 database).

**Figure 12 jintelligence-13-00045-f012:**
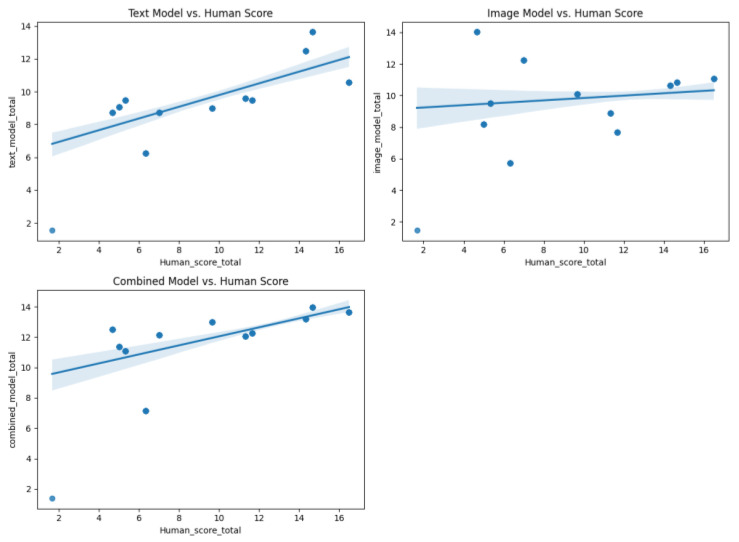
Scatter plot of different model sum scores against human scores (100 database).

## Data Availability

We generated a public archive OSF data link on 10 January 2025: https://osf.io/g64fv/.

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
