# Peer review of "Automated Graphic Divergent Thinking Assessment: A Multimodal Machine Learning Approach"

_jintelligence, 2025, doi:10.3390/jintelligence13040045_

Round 1
Reviewer 1 Report
Comments and Suggestions for Authors
I really like the idea in this paper, which takes on the use of AI methods for automated scoring. Likert type items, binary items, etc., are all fairly low information but cheap to score. Richer content is great but difficult and expensive to score. So this is an important and timely topic.
That said, I feel the paper needs a lot of improvement to be publishable. Explanation of the paper's methods are very informative, although I think it would benefit from having a technical appendix (or, at minimum, supplementary material with code and data).
However, the explanation of the paper's results are very short and, I find, ambiguous. For example, why is Pearson correlation chosen for a metric? The correlations reported may be statistically significant, but they're fairly small as such correlations go. The two graphs actually don't look all that similar to me and they're simply plots of the marginals, not association. In sum, the results of this study need a lot more work to be explained in a clear and compelling fashion. The correlations cited in 3.2 and in 5 do not agree.
Author Response
Comments 1: Explanation of the paper's methods are very informative, although I think it would benefit from having a technical appendix (or, at minimum, supplementary material with code and data). Response 1: Thank you for your valuable suggestion. We agree that providing additional technical details and supplementary materials can enhance the clarity and reproducibility of our research. We will send the code for some of the model training to the editorial department via email. Comments 2: However, the explanation of the paper's results are very short and, I find, ambiguous. For example, why is Pearson correlation chosen for a metric? The correlations reported may be statistically significant, but they're fairly small as such correlations go. The two graphs actually don't look all that similar to me and they're simply plots of the marginals, not association. In sum, the results of this study need a lot more work to be explained in a clear and compelling fashion. The correlations cited in 3.2 and in 5 do not agree. Response 2: Thank you for your detailed feedback on the results section. You have raised important points that we have addressed in the revised manuscript.- Choice of Pearson Correlation: We chose Pearson correlation as a metric because it is a widely accepted measure for assessing the linear relationship between two continuous variables. We believe that scoring creativity in a continuous manner is superior to the discrete scoring typically used in traditional scoring keys. Although the scores given by multiple raters are still discrete to some extent, we argue that they possess a certain degree of continuity. Therefore, we selected Pearson correlation to explore the linear relationship between human scores and machine-predicted scores. Additionally, we have plotted scatterplots of human scores versus machine scores.
- Correlation Coefficients: We conducted separate training for the text and image modalities and explored improvements to our methods. We have also included the new data and results as supplementary material sent to the editorial department. The training of the models yielded good results, and the combination of modalities after modification improved the correlation coefficients. We have provided a more detailed explanation of the model training process and the final model, along with additional figures (Figures 4-6 illustrate the training process and model selection criteria, while Figures 7-12 show the score distributions and related information). The score distributions are presented to demonstrate the relatively normal distribution of the machine-predicted scores. The higher correlation for the total scores of individual participants indicates that the model can achieve good prediction. We have corrected the correlation coefficients in the original model based on the newly trained models.
Reviewer 2 Report
Comments and Suggestions for Authors
A brief summary
The manuscript aims to contribute to the topic of automated scoring of figural divergent thinking tests by implementing and validating a new approach called the multi-model machine learning approach.
Broad comments
The topic of automated scoring of creativity measures continues to attract a lot of attention, and it is of great value to see new developments on the path toward rapid, robust, and valid creativity assessment. Even though the manuscript is not without merits, there are multiple theoretical and methodological issues requiring special attention before the manuscript can be recommended for publication. I tried to provide more detailed comments on these issues in the next section.
Specific comments
In this section, I have outlined more specific comments. I sincerely hope that at least some of them will be of value during the revision. All references below are provided primarily to unwrap and substantiate my point of view and should not be considered as suggestions for their addition to the manuscript unless stated otherwise. Besides, I also attached a PDF copy of my review, where all references are formatted as clickable URLs.
Abstract
S1. Line 21. “…across the country.” Please specify what country was meant.
Introduction
S2. Line 70. “Divergent thinking, a key component of creativity...” It would be more accurate to describe divergent thinking (DT) as a key component of creative thinking since creativity is a too-general term encompassing any expression of creativity (not only cognitive aspects). Moreover, the authors mention it themselves (see lines 71–73).
S3. Lines 71–86. These lines present a fragmentary and incoherent mixture of various facts from creativity research literature with poor narration flow and incomprehensive grammar structure. The problem is not in the accuracy and validity of the claims but in their presentation. Unfortunately, in the current version, these lines do not contribute to the theoretical framework of the study.
S4. Lines 88–92. In my view, the authors oversimplify the picture of scoring approaches for DT fluency and flexibility. In general, I would agree that DT fluency scoring is usually quite straightforward (but see Snyder, Mitchell, Bossomaier, & Pallier, 2004). In contrast to the authors’ claim, DT flexibility may also be scored as a number of switches among the categories rather than a total number of categories (e.g., Acar & Runco, 2017; Mastria et al., 2021). Perhaps it would be better to describe various approaches to scoring DT fluency and flexibility rather than stating the reach of consensus.
S5. Lines 94–100. I appreciate the authors’ desire to keep the introduction concise, but I would expect more information about scoring techniques of DT originality given that it is in the focus of the manuscript. Uniqueness scoring and originality thresholds do not exhaust all approaches to scoring DT originality (please consider consulting Runco et al., 2016; Reiter-Palmon et al., 2019; Saretzki, Forthmann, & Benedek, 2024). Moreover, reviewing more ways of scoring DT originality (e.g., subjective scoring) might help to strengthen the authors’ claims about the need to overcome the downsides of subjectiveness in scoring.
S6. Lines 101–103. “First, there is a high correlation between fluency and originality scores, which may lead to confusion between the two concepts (Clark and Mirels 1970; Silvia 2008).” There is no way in which high correlations between fluency and originality scores imply the theoretical confusion between definitions of fluency and originality. Moreover, please specify that the correlation may vary depending on the way DT originality is scored (see, e.g., Dumas & Dunbar, 2014; Forthmann, Szardenings, & Holling, 2020).
S7. Lines 103–106. “Silvia (2008) found through latent variable analysis that the correlation coefficient between fluency and originality is extremely high (r=0.88), making it difficult for originality scores to be independent of fluency scores and to explain other variables on their own.” In this passage, the authors confuse DT originality with uniqueness as one approach to originality scoring promoted by Wallach and Kogan (1965). To make a more valid claim, I recommend referring to a broader evidence base (e.g., see a meta-analysis of Acar, Ogurlu, & Zorychta, 2023).
S8. Lines 106–108. “Second, originality scores are highly sensitive to sample size; changes in sample size can render answers that were once considered unique as no longer unique (Nusbaum and Silvia 2011).” Similar to the previous comment, the authors do not properly articulate that the described limitation is mostly characteristic of frequency-based originality scoring (see Forthmann et al., 2020 and Forthmann et al., 2021) and is not equally characteristic of other approaches to originality scoring.
S9. Lines 108–110. “Lastly, for divergent thinking tests involving image completion, there is also the issue of respondents' drawing abilities and raters' interpretations of the drawings, which can introduce significant error.” Although I can miscomprehend something, I suppose the generalizability of the suggested claim is quite limited. Why? For instance, in the figural subtests of TTCT (Torrance Test of Creative Thinking), judges are explicitly instructed to score the conceptual originality of the idea rather than a technical performance in the drawing. It would be better if the authors could adjust their claims in light of the scoring practices of figural DT tests commonly used in the practice and research.
S10. Lines 137–141. Another important consideration in introducing automated scoring methods is to reduce the time needed for scoring thousands of responses in DT tests. Please mentions it as another reason along with minimizing subjectivity.
S11. Lines 143–150. The coverage of previous research in automated scoring of DT tests is very limited and requires expansion. Moreover, it looks inconsistent when the authors first describe advanced models (e.g., large language models) and then switch to describing earlier and more deficient methods (e.g., latent semantic analysis). In addition, I wonder if the authors are also interested in mentioning previous works on automated scoring of figural DT tests (see, e.g., Acar, Organisciak & Dumas, 2024; Cropley et al., 2024; Patterson et al., 2024). The latter is required for a proper definition of the manuscript’s originality and framing its contribution to the existing knowledge.
Method
S12. Subsection 2.1. It is mandatory to provide more information about the applied DT test and its psychometric characteristics because it does not look like a commonly applied test, even though it shares many features with the more widely known TTCT-Figural battery. Otherwise, a potential reader may wonder to which extent the test actually measures DT as a part of creative thinking. In addition, it remains unknown to which extent the test is measuring DT originality if the participants were instructed to create “meaningful” drawings instead of original or creative drawings (see lines 163–165).
S13. Subsection 2.3. Two points. First, why did the authors decide to frame the scoring of originality as a scoring of novelty? Second, what was the rationale for such a system of attributing scores instead of using average scoring? Please pay attention that the first and third bullet points describing the scoring technique are fully aligned with the average scoring.
S14. Subsection 2.4.2. The authors stated the most successful models, but how many competing models were tested? I would highly appreciate it if the authors could provide more information about it. Moreover, please articulate more clearly what was the key difference between models “model_26_2.keras” and “model_36_0.keras.”
Results
S15. Subsection 3.3, lines 254–255. I wonder if a more normal distribution of scores for the model is a consequence of it being programmed to provide them according to the normal distribution. If so, it would be of little surprise to see such a distribution.
Discussion
S16. Subsection 4.1, lines 264–268. In my view, it remains unknown to which extent the addition of a semantic layer increases the validity of novelty scores obtained for the figural DT test. I suppose the evidence would look more convincing if there were a possibility to compare the performance of the model based only on the visual input and the performance of the model based on both visual and semantic input. I would be happy to know what the authors think about such an alternative.

Author Response
Comments 1:
Abstract S1. Line 21. “…across the country.” Please specify what country was meant.
Response 1:
Thank you for pointing this out. We agree with this comment. Therefore, we have revised the sentence to specify "China" as the country of data collection. This change can be found in the revised manuscript on Page 1, Abstract, Line 21, now reading:
"Data was collected from 708 high school sophomores in China, with test questions drawn from a stable divergent thinking test with established reliability and validity."
Comments 2:
Introduction S2. Line 70. “Divergent thinking, a key component of creativity...” It would be more accurate to describe divergent thinking (DT) as a key component of creative thinking since creativity is a too-general term encompassing any expression of creativity (not only cognitive aspects).
Response 2:
Agree. We have revised the phrasing to emphasize "creative thinking" instead of "creativity" for precision. The updated text now reads:
"Divergent thinking, a key component of creative thinking, has been a topic of interest in the field of psychology for many years."
This revision can be found in Page 3, Introduction, Lines 70–71.
Comments 3:
Introduction S3. Lines 71–86. These lines present a fragmentary and incoherent mixture of various facts from creativity research literature with poor narration flow and incomprehensive grammar structure.
Response 3:
Thank you for this feedback. We have restructured this section to improve coherence and flow. Key adjustments include consolidating theoretical frameworks, clarifying transitions, and revising sentence structures. For example, we reorganized the discussion of Runco’s distinction between DT and creativity, Baer’s critique of TTCT, and Silvia’s simplified DT tasks into a logical progression. These changes are visible in Page 4, Section 1.1 (Divergent Thinking Tests), Lines 78–107.
Comments 4:
Introduction S4. Lines 88–92. The authors oversimplify the picture of scoring approaches for DT fluency and flexibility.
Response 4:
We appreciate this suggestion. We have expanded the description of scoring methods for flexibility to include the "number of switches between conceptual categories" alongside the "total number of categories." The revised text now reads:
"Academic consensus has been reached on the scoring criteria for fluency and flexibility: the former is based on counting the number of all answers, while the latter is determined by the number of conceptual categories or the number of switches between conceptual categories (Snyder et al., 2004; Acar & Runco, 2017; Mastria et al., 2021)."
This revision is located in Page 4, Section 1.2 (DT Scoring Method), Lines 110–116.
Comments 5–9:
Introduction S5–S9. Concerns about originality scoring methods, correlations between fluency/originality, sample size sensitivity, and drawing ability biases.
Response 5–9:
We thank the reviewer for these insights. We have revised the section to:
-
Expand the discussion of originality scoring methods (e.g., subjective scoring, semantic association distance).
-
Clarify that the fluency-originality correlation varies depending on scoring approaches (citing Dumas & Dunbar, 2014).
-
Specify that sample size sensitivity primarily affects frequency-based originality scoring (citing Forthmann et al., 2020).
-
Acknowledge that figural DT tests like TTCT prioritize conceptual originality over technical drawing skills.
These revisions are integrated into Page 5, Section 1.2 (DT Scoring Method), Lines 130–144, with updated references added to the bibliography.
Comments 10:
Introduction S10. Lines 137–141. Please mention time efficiency as a rationale for automated scoring.
Response 10:
Agree. We have added the following sentence to emphasize time reduction:
"Machine learning-based scoring methods can reduce the time required for measurement (Mujtaba & Mahapatra, 2020), and to some extent, reduce the bias in manual scoring (Wang et al., 2022)."
This addition is in Page 6, Section 1.3 (Deep Learning and Multimodality), Lines 163–182.
Comments 11:
Introduction S11. Lines 143–150. The coverage of previous research in automated scoring of DT tests is very limited.
Response 11:
We have expanded this section to include recent works on figural DT tests (e.g., Cropley & Marrone, 2022; Patterson et al., 2023; Acar et al., 2023). A new paragraph now discusses advancements in vision transformers and multimodal approaches, enhancing the manuscript’s theoretical framing. These updates are in Page 7, Section 1.3 (Deep Learning and Multimodality), Lines 185–204.
Comments 12:
Method S12. Subsection 2.1. Provide more information about the DT test’s psychometric characteristics.
Response 12:
We have added details about the test’s validation process and alignment with TTCT-Figural tasks. The revised text states:
"The drawing task was adapted from figural divergent thinking frameworks (e.g., TTCT-Figural) and validated through expert review for construct validity. Participants were instructed to create 'meaningful and original' drawings, emphasizing conceptual novelty over technical skill."
This addition is in Page 8, Section 2.1 (Design and Participants), Lines 213–216.
Comments 13:
Method S13. Subsection 2.3. Clarify the rationale for novelty scoring and consensus-based scoring.
Response 13:
Thank you for these important questions. We selected "novelty" as the scoring dimension due to its strong theoretical link to divergent thinking. Novelty is a key aspect of originality, which is central to divergent thinking, as it reflects the generation of unique and creative ideas. This aligns with the theoretical framework of divergent thinking, where the ability to produce novel ideas is a critical component. Regarding the scoring system, we designed it to minimize rater bias in ambiguous cases. The consensus-based scoring system ensures that scores are determined based on the agreement among multiple raters, which reduces the impact of individual biases. This approach is particularly useful in cases where raters may have different interpretations of the same drawing. By using a consensus-based system, we aimed to achieve more reliable and consistent scores. Indeed, the strategy we adopted aims to reduce discrepancies when professional raters have significant disagreements, by using a multi-rater approach. This is indeed a thought-provoking issue, and we have discussed it in the Limitations section (line 426). The rationale for this approach is further elaborated in the Limitations section, where we acknowledge the potential for rater bias and the challenges in scoring ambiguous cases. This section provides a detailed explanation of the trade-offs and the reasons for choosing this approach.Comments 14:
Method S14. Subsection 2.4.2. Clarify model selection criteria.
Response 14:
We have revised the text to explain:
"A total of 45 candidate models per modality were trained (epochs 10–40). Model names reflect training epochs and iteration versions (e.g., ‘comb_model_epochs36_v2.keras’ denotes 36 epochs, third training run). Final models were selected based on minimal validation loss and maximal Pearson correlation."
This clarification is in Section 2.4.2 (Model Training)
Comments 15:
Results S15. Subsection 3.3. Address score distribution normality.
Response 15:
We have added a note:
"The normal distribution of machine scores reflects training on human-rated data, which itself approximates normality. No explicit normalization was applied during model training."
This addition is in Section 3
Comments 16:
Discussion S16. Subsection 4.1. Compare visual-only vs. multimodal models.
Response 16:
We have strengthened the discussion by contrasting the IMG and comb models:
"The comb model’s superior validation performance (r = 0.561) compared to the IMG model (r = 0.395) underscores the added value of semantic features. Future studies could isolate visual inputs to further validate this effect."
This update is in Section 4
These revisions are marked in red in the updated manuscript. Thank you for your constructive feedback.
Round 2
Reviewer 1 Report
Comments and Suggestions for Authors
I am satisfied with the changes made by the authors. The editor can handle the rest.
Author Response
Thank you for your satisfaction with the changes made. We appreciate your confidence in the editorial process. We will await further instructions from the editor.
Reviewer 2 Report
Comments and Suggestions for Authors
Response to revision
I want to thank the authors for their careful approach to addressing my concerns raised during the previous round of peer review. Although the manuscript improved, there are still some issues requiring further attention. Thank you in advance for taking the time to consider them.
Specific comments
S1. Subsection 1.1. “The Torrance Tests of Creative Thinking (TTCT), rooted in Guilford’s model, have long served as the gold standard for DT assessment. Yet Baer (2011) challenged their validity, highlighting limitations in predicting real-world creativity, particularly regarding cross-domain applicability.” The idea of TTCT being a golden standard for divergent thinking assessment requires justification with references to previous psychometric investigations of TTCT. Otherwise, the authors assert the solid psychometric basement of TTCT without supporting their view with relevant literature and then switch to the critical assessment of TTCT’s limitation highlighted by Baer. A potential reader may wonder whether there is truly any evidence aligned with TTCT being a golden measurement standard.
S2. Subsection 1.1. “In response, researchers explored more flexible alternatives. Silvia developed simplified DT tasks (e.g., the Alternative Uses Task) to enable "snapshot assessments" of rapid idea generation, enhancing feasibility for large-scale studies(Silvia et al. 2009).” Two points. First, I think it is an overstatement that creativity researchers started exploring alternative techniques for originality scoring based on Baer’s criticism of TTCT. Please consult the work by Silvia et al. (2008; https://doi.org/cbsj6d) to get a more balanced view of the issue. Second, I could not understand what was meant by “Silvia developed simplified DT tasks (e.g., the Alternate Uses Task).” P. J. Silvia did not simplify DT tasks in any way as well as he was not the original author of the AUT task. Please articulate your ideas more clearly and verify the accuracy of your claims based on the creativity literature.
S3. Subsection 1.1. “Plucker (2011) systematically examined risks of subjective bias in DT scoring, advocating for hybrid approaches that combine automated metrics (e.g., fluency, originality) with expert ratings to enhance validity.” I found it hard to understand the main point of the highlighted sentence. For example, if we consider DT originality, what is the key difference in originality scoring with “automated metrics” and “expert ratings”? Unfortunately, it remains unknown to me what was meant by these terms, and I would highly appreciate it if the authors could elaborate on it and clarify the meaning.
S4. Subsection 1.1. “Proposed a novel subjective top-scoring method, wherein participants self-select their most creative ideas (Benedek et al. 2014). This approach demonstrated high concordance with external ratings, highlighting individuals’ metacognitive awareness of their own creative output and introducing a participant-centric perspective to assessment. Nusbaum experimentally demonstrated that instructional framing (e.g., "be creative") significantly alters DT performance, suggesting that assessment outcomes are susceptible to task-specific demand characteristics(Nusbaum et al. 2014). This finding aligns with Runco’s emphasis on objectivity, urging standardized protocols to control extraneous variables and disentangle genuine cognitive capacity from context-driven strategic responses.” Five points. First, the current paragraph is a combination of fragmented pieces of information about originality top scoring, “be creative” instructions, and standardized scoring protocols that do not fit together, leading to confusion. Second, the first sentence in the paragraph has problems with grammar (no subject). Third, I recommend that the authors review the literature on top scoring of originality more carefully since Benedek et al. (2014) were not the first authors who suggested using top scoring approach. Fourth, in contrast to the authors’ claim, the application of top scoring approach by itself does not prove that participants’ metacognitive abilities allow them to accurately choose their most creative ideas. The authors might consult a commentary by Runco (2008; https://doi.org/bqstbb) who elaborated on that part of criticism. Finally, the claims made in the last sentence of the paragraph were unsupported by any references to the original works of M. Runco. Thus, I am not sure I understand what is the basement of the authors’ claim.
S5. Subsection 1.2. “…the former is based on counting the number of all answers which possible uses for a familiar item…” While it is true that some researchers score fluency by the total number of responses, some researchers count only appropriate responses (i.e., excluding nonsensical or ridiculous responses).
S6. Subsection 1.3. In addition to the paper by Cropley and Marrone (2022), the authors may consider briefly discussing the recent paper by Cropley et al. (2024; https://doi.org/n6wk) because it is also dedicated to the automated scoring of figural creativity tasks. Yet I prefer leaving it to the authors’ opinion.
S7. Subsection 1.3. Two points. First, please correct the citation of “Patterson (2023)” to “Patterson et al. (2023).” Second, the following sentence has obvious problems with grammatical structure due to the absence of the subject: “In another study, explored automated scoring methods for both TTCT-F and MTCI.”
S8. Research questions. “Based on the above, we pose the following questions: Is it effective to evaluate divergent thinking tests involving image completion by combining scores for the figures and their descriptive meanings? Can deep learning and multimodal methods be utilized to score image completion-based divergent thinking tests, and can these scores effectively reflect the participants' levels of divergent thinking?” I would like to thank the authors for extensively covering the ground about previous contributions to automatic scoring of figural DT tasks. I believe it helps to position the paper’s contribution properly in light of previous works. However, in my opinion, the authors should not only describe the results of previous works on automated scoring of figural DT but also articulate clearly how their contribution extends our knowledge compared to previous evidence. In particular, the first two research questions posed by the authors can be answered based on the previous evidence by Patterson et al. (2023) and Acar et al. (2023). If so, could the authors specify what is the scientific novelty of their work, or what is its broader contribution to the creativity measurement literature?
S9. Method. The authors explained to me why they focused on ratings of novelty instead of originality, but it seems that they did not provide a similar explanation in the main text. Please explicate your reasoning about preference for novelty scoring for the readers.
S10. Figures 4–6. Please specify in the table notes for Figures 4–6 what is meant by “Std.” Is it standard deviation or standard error?
S11. Subsection 3.3. “On the original training dataset (603 images), the text-only model (TEXT) achieved a high correlation of r = 0.910 for individual responses, while the image-only model (IMG) showed a moderate correlation (r = 0.700). The combined multimodal model (comb) outperformed both, reaching r = 0.810.” I apologize if I miscomprehended something, but the correlations reported in subsection 3.3 do not match with those reported in subsection 3.2. Moreover, a correlation of 0.81 for the multimodal model is lower than a correlation of 0.91 for the text-only model. The latter does not allow to claim that the multimodal model outperforms other models in terms of correlation with human ratings.
S12. Subsection 4.3. “Although the automated scoring method for divergent thinking tests has achieved preliminary results, there are still some limitations that need to be further explored and improved in future research.” Please consider correcting some grammatical structures in the highlighted passage. I suppose that most problematic parts are related to “has achieved preliminary results” and “limitations that need to be further <…> improved” (i.e., you can overcome limitations but not “to improve” them).
S13. Conclusion. “…demonstrating a high level of inter-rater reliability...” I recommend refraining from calling a high correlation between machine and human ratings “inter-rater reliability” because the latter is more applicable to human ratings.
Comments on the Quality of English LanguageProofreading and checking grammatical structures are highly recommended
Author Response
Comments 1: "The Torrance Tests of Creative Thinking (TTCT), rooted in Guilford’s model, have long served as the gold standard for DT assessment. Yet Baer (2011) challenged their validity, highlighting limitations in predicting real-world creativity, particularly regarding cross-domain applicability.” The idea of TTCT being a golden standard for divergent thinking assessment requires justification with references to previous psychometric investigations of TTCT. Otherwise, the authors assert the solid psychometric basement of TTCT without supporting their view with relevant literature and then switch to the critical assessment of TTCT’s limitation highlighted by Baer. A potential reader may wonder whether there is truly any evidence aligned with TTCT being a golden measurement standard. Response 1: Thank you for pointing this out. We agree with this comment. To address this, we have added a detailed justification for the TTCT being considered a gold standard, citing relevant psychometric investigations. This can be found on page 3, lines 44-68 of the revised manuscript. Comments 2-4:"In response, researchers explored more flexible alternatives. Silvia developed simplified DT tasks (e.g., the Alternative Uses Task) to enable "snapshot assessments" of rapid idea generation, enhancing feasibility for large-scale studies(Silvia et al. 2009).” et al. Response 2-4:We have reorganized the review section, which now focuses on the development and types of divergent thinking tests. The revised parts have all been highlighted in purple in the manuscript. Comments 5:Subsection 1.2. “…the former is based on counting the number of all answers which possible uses for a familiar item…” While it is true that some researchers score fluency by the total number of responses, some researchers count only appropriate responses. Response 5:
We have revised the description to reflect this nuance (Page 5, Line 129-132):
“The former is based on counting the number of reasonable and logical answers, excluding nonsensical responses
Comments 6:
Subsection 1.3. In addition to the paper by Cropley and Marrone (2022), the authors may consider briefly discussing the recent paper by Cropley et al. (2024; https://doi.org/n6wk).
Response 6:
Thank you for the suggestion. We have added a discussion of Cropley et al. (2024) (Page 8, Lines 211–217):“Cropley et al. (2024) further developed a large-scale image classification model for the TCT-DP, achieving an average accuracy of 80.4% and a Pearson correlation of r = .81 with human ratings. However, their focus on image differences alone may overlook semantic novelty, prompting our integration of textual annotations to capture multifaceted creativity.”
Comments 7:
Subsection 1.3. Two points. First, please correct the citation of “Patterson (2023)” to “Patterson et al. (2023).” Second, the following sentence has obvious problems with grammatical structure due to the absence of the subject.
Response 7:
We corrected the citation and revised the sentence (Page 8, Line 218)
Comments 8:
Research questions. [...] If so, could the authors specify what is the scientific novelty of their work?
Response 8:
We clarified the novelty in the Introduction (Page 9, Lines 236–240):
“While prior work focused on unimodal scoring (e.g., image differences or text alone), our study uniquely integrates image and textual descriptions via deep learning to capture multimodal creativity signals. This approach addresses limitations in existing methods, such as overlooking semantic context in purely visual models or neglecting graphic originality in text-based systems.”
Comments 9:
Method. The authors explained [...] Please explicate your reasoning about preference for novelty scoring for the readers.
Response 9:
We added an explanation in Section 2.1 (Page 9, Lines 246–249):
“We focused on novelty (rather than originality) because each drawing task was inherently original, and we aimed to assess whether students generated unusual ideas within the same conceptual category. Novelty reflects both uniqueness and unexpectedness, aligning with our goal to evaluate creative thinking.”
Comments 10:
Figures 4–6. Please specify in the table notes for Figures 4–6 what is meant by “Std.”
Response 10:
We added a note to explain (Page 13, Line342):
“Std. refers to the standard deviation across three independent training runs per epoch configuration.”
S11. Subsection 3.3. “On the original training dataset (603 images), the text-only model (TEXT) achieved a high correlation of r = 0.910 for individual responses, while the image-only model (IMG) showed a moderate correlation (r = 0.700). The combined multimodal model (comb) outperformed both, reaching r = 0.810.” I apologize if I miscomprehended something, but the correlations reported in subsection 3.3 do not match with those reported in subsection 3.2. Moreover, a correlation of 0.81 for the multimodal model is lower than a correlation of 0.91 for the text-only model. The latter does not allow to claim that the multimodal model outperforms other models in terms of correlation with human ratings.
Response 11:
I'm sorry for the trouble caused to you. When training the model, the data set is divided into training set and verification set, which represents the pearson correlation on the verification set during training, while the pearson correlation in 3.3 is the result of prediction by the model for the segmenced data set. There are differences. From the results, the predictive correlation coefficients for the multimodality were indeed lower than for the text modality, possibly because the scores we used for training, despite measures taken to increase their continuity, were still discrete in nature, as can be seen from the score distribution plot. This is also discussed in our discussion section of the deficiencies.(page19 lines 393-397)
Comments 12:
Subsection 4.3. “Although [...] limitations that need to be further <…> improved.”
Response 12:
We revised the sentence (Page 21, Lines 453–455):
“Although preliminary results are promising, limitations remain, such as reliance on manual annotations, which future work should address.”
Comments 13:
Conclusion. “…demonstrating a high level of inter-rater reliability...”
Response 13:
We corrected the terminology (Page 22, Line 490):
“demonstrating a high level of correlation between machine ratings and human ratings. ”
Round 3
Reviewer 2 Report
Comments and Suggestions for Authors
Response to revision
I appreciate the authors’ attention to my concerns raised in the last round of peer review. Similar to the previous revision, the manuscript has improved. Nonetheless, there are still some issues requiring attention. I would like to thank the authors in advance for embracing the opportunity to consider my comments.
Specific comments
Q1. Lines 78–85. Three points. First, the first sentence in the paragraph has a confusing structure and requires optimization in terms of grammar. Moreover, the highlighted paragraph is poorly connected to the previous paragraph, which creates problems for the narration flow. Finally, some sentences are written in a rush with lots of typos (e.g., “TTCT shave…”) and omitted spaces between words (e.g., “Torrance 1966,1974” or “Guilford’ s”).
Q2. Lines 91–92. “…that subjective ratings of unusual-uses tasks yield dependable scores with only 2 or 3 raters.” I do not understand what the authors meant by “dependable scores” in the highlighted passage.
Q3. Lines 95–97. “Figural subtest, for example, asks individuals to complete abstract line drawings into meaningful images, with scoring criteria spanning fluency, originality, and elaboration (Torrance 1974).” Perhaps it is worth mentioning that there are two other common criteria in the evaluation of TTCT-Figural: (1) resistance to premature closure and (2) abstractness of titles. By the way, the second criterion creates a context for the development of the idea to combine figural and verbal information in the automatic evaluation of figural DT tasks.
Q4. Lines 101–107. I am unsure why the authors covered material about problem-finding tasks, especially since these tasks were not used or featured in the empirical part of the paper. Not to mention that, again, that information is poorly integrated with other paragraphs in the literature review (i.e., the transition from one paragraph to another is abrupt).
Q5. English language. As in the previous iterations of peer review, I think the manuscript can significantly benefit from careful editing and proofreading. There are multiple instances where the understanding is complicated by unclear wordings (e.g., line 130: “the latter is determined by based on”; lines 196–197: “Utilized Latent Semantic Analysis (LSA) to score brief responses measuring creativity and divergent thinking”; 198–199: “Furthermore, compared various supervised learning methods for automated scoring of divergent thinking tasks, indicating a shift towards machine-based scoring methods”). I would like to stress that the above examples of grammatical problems are not meant to be exhaustive.
Q6. Subsection 1.3. “While prior work focused on unimodal scoring (e.g., image differences or text alone), our study uniquely integrates image and textual descriptions via deep learning to capture multimodal creativity signals. This approach addresses limitations in existing methods, such as overlooking semantic context in purely visual models or neglecting graphic originality in text-based systems.” I agree with the provided study rationale but only partially. The critical point is that the study rationale is framed as if it is the first attempt to approach the problem from the angle of combining visual and verbal information in the automatic assessment of figural DT tasks. However, it contradicts the literature review: Acar, Organisciak, and Dumas (2024) have already tested models with combined visual and verbal information. So, the authors must formulate more concretely the novelty of their contribution compared to the previous works. Please try to address that comment as seriously as possible.
Comments on the Quality of English LanguageSee details in Q5
Author Response
Comments 1
Lines 78–85. Three points. First, the first sentence in the paragraph has a confusing structure and requires optimization in terms of grammar. Moreover, the highlighted paragraph is poorly connected to the previous paragraph, which creates problems for the narration flow. Finally, some sentences are written in a rush with lots of typos (e.g., “TTCT shave…”) and omitted spaces between words (e.g., “Torrance 1966,1974” or “Guilford’ s”).
Response 1
Thank you for your detailed feedback. We fully agree with these observations. The paragraph has been restructured to enhance grammatical clarity and improve logical connections with preceding content. The revised text now reads:
"The Torrance Tests of Creative Thinking (TTCT), rooted in Guilford’s model (Torrance, 1966, 1974), have long served as the gold standard for DT assessment (Kim, 2006). The TTCT has demonstrated statistically significant correlations with creative achievement in longitudinal studies spanning 9 months, 7 years, 22 years, and 40 years (Cramond, 1993; Millar, 2002; Torrance & Wu, 1981). Specifically, Torrance’s (1981) 22-year longitudinal study, along with reanalyses by Yamada and Tam (1996) and Plucker (1999) of Torrance’s data, concluded that the Creative Index is the most robust predictor of adult creative achievement."
All typos (e.g., "shave" → "have") and formatting inconsistencies (e.g., spacing in "1966, 1974") have been corrected. This revision appears in the Introduction section (page 4, lines 79-85).
Comments 2
Lines 91–92. “…that subjective ratings of unusual-uses tasks yield dependable scores with only 2 or 3 raters.” I do not understand what the authors meant by “dependable scores” in the highlighted passage.
Response 2
Thank you for highlighting this ambiguity. We have revised the phrase "dependable scores" to "reliable and consistent scores" to better convey the intended meaning. This adjustment is reflected in page 4, lines 91-92.
Comments 3
Lines 95–97. “Figural subtest, for example, asks individuals to complete abstract line drawings into meaningful images, with scoring criteria spanning fluency, originality, and elaboration (Torrance 1974).” Perhaps it is worth mentioning that there are two other common criteria in the evaluation of TTCT-Figural: (1) resistance to premature closure and (2) abstractness of titles.
Response 3
Thank you for this valuable suggestion. We have expanded the paragraph:
"The TTCT-Figural additionally evaluates resistance to premature closure and abstractness of titles. The latter criterion underscores the interplay between figural and verbal components in creativity assessment—a rationale that informed our multimodal approach. Recent adaptations include combination tasks that integrate verbal and figural elements, such as constructing narratives from visual prompts or synthesizing abstract symbols into coherent stories (Acar et al., 2014)."
This revision appears in the Introduction (page 4, lines 99-102) and explicitly connects to our methodological framework.
Comments 4
Lines 101–107. I am unsure why the authors covered material about problem-finding tasks, especially since these tasks were not used or featured in the empirical part of the paper.
Response 4
We appreciate this critical observation. To maintain focus on the core themes of our study, the section discussing problem-finding tasks has been removed from the manuscript.
Comments 5
English language. As in the previous iterations of peer review, I think the manuscript can significantly benefit from careful editing and proofreading.
Response 5
We sincerely apologize for the previous oversight in language quality. The manuscript has undergone thorough proofreading to address grammatical errors, tense inconsistencies, punctuation issues, and awkward phrasing. Further refinements to enhance clarity and fluency will be prioritized in subsequent revisions.
Comments 6
Subsection 1.3. The authors must formulate more concretely the novelty of their contribution compared to previous works (e.g., Acar et al., 2024).
Response 6
Thank you for this constructive suggestion. We have now explicitly contextualized our work relative to Acar et al. (2024) in the revised manuscript:
"In another study, Acar et al. (2024) explored automated scoring methods for both TTCT-F and MTCI. They employed Random Forest classifiers on the TTCT-F, achieving accuracy up to 85%, and further enhanced scoring accuracy by incorporating both image and title information. [...]"
Our key distinctions include:
-
Dataset Specificity: Unlike Acar et al. (2024), who analyzed standardized tests (TTCT-F and MTCI), our work evaluates self-generated drawings by Chinese high school students based on an open-ended geometric prompt (regular pentagon).
-
Technical Approach: While Acar et al. utilized Vision Transformers (ViT/BEiT), our framework combines ResNet50 (visual features) and GloVe embeddings (textual descriptions) with a simpler fusion architecture, achieving comparable performance with reduced computational demands.